# Soil Erosion and Land Degradation on Trail Systems in Mountainous Areas: Two Case Studies from South-East Brazil

**Luana Rangel [1,*], Maria do Carmo Jorge [1], Antonio Guerra [1] and Michael Fullen [2]**

[1]   Department of Geography, Federal University of Rio de Janeiro, Rio de Janeiro 21941-909, Brazil
[2]   Faculty of Science and Engineering, University of Wolverhampton, Wolverhampton WV1 1LY, UK
*   Correspondence: luarangel24@gmail.com

**Abstract:** This paper addresses the role of soil erosion and mass movements on mountainous trails due to human trampling on steep slopes. This is the case of several trails situated on forested areas in South-East Brazil, even those located in protected areas. Two methods were used to achieve the research objectives. Firstly, analyses of microtopography using erosion bridges, which was monitored four times on *Caixa D'Aço* natural pool trails in *Serra da Bocaina* National Park. Secondly, disturbed and undisturbed soil samples were collected at 0–10 cm depth at four sites on *Água Branca* trail in *Serra do Mar* State Park. Using this methodology, we assessed soil degradation in two different humid tropical environments. Generally, trampling combined with deficient trail management, play important roles in degrading soils in both areas. Bioengineering techniques should be used to recuperate these trails, which are used by tourists and local residents. We hope this research work may contribute towards improved management in Brazilian protected areas.

**Keywords:** soil conservation; land conservation; soil erosion; trails; protected areas

---

## 1. Introduction

Land degradation occurs globally, being more dramatic in tropical areas, where erosive rainfall regimes often cause severe soil erosion [1–5]. Forms of land degradation include mass movements, soil erosion, acidification, salinization and desertification.

Climate change plays important roles in these degradation processes, especially in tropical countries, where soil degradation exacerbates environmental and social problems [2,3,5–7]. Land degradation "*has emerged as a serious problem during the last few decades, consequently, soil fertility has declined considerably in many parts of the world due to intensive agriculture, over-grazing, water pollution, increasing use of fertilizers and pesticides, salinization, deforestation and accumulation of non-biodegradable waste*" [8].

The total land area of the Earth is estimated at 148,940,000 km$^2$, with arable land forming an estimated 13,958,000 km$^2$. Therefore, arable land constitutes <10% of the land area of the Earth [9] Total agricultural area has been estimated at 4.889 billion hectares, consisting of arable land (28%), permanent crops (3%) and meadows and pastures (69%) FAO [9].

Soil erosion and land degradation destroy much of the environment, thus damaging both biotic and social systems. Trails are one of the landscape components affected by land degradation. These are used both by local residents and by tourists, especially where geotourism is being adopted. This economic activity, if not well managed, may cause serious problems affecting both the trails and their environments. Therefore, this study addresses soil erosion and land degradation, in *Paraty* Municipality (*Rio de Janeiro* State) and *Ubatuba* Municipality (*São Paulo* State), considering trails compacted by trampling by both tourists and residents [10].

Trails are integral components of cultural landscapes and have long been used as communication corridors. In recent decades, geotourism has increased in importance [10–13]. Trails have multiple functions, including accessing natural environments, nature contemplation, recreation and sporting activities [14–18].

Trampling can degrade trails [17–19] and trail mismanagement can exacerbate land degradation (i.e., decreased soil quality and capacity as an environmental regulator [20–23]. Many studies have investigated soil degradation on trails. Trampling may cause physical degradation, including soil compaction and erosion, chemical and biological depletion, with associated losses of nutrients and soil organic matter (SOM) and decreased soil faunal activity [14–16,19,24–26].

It is important that land planners assess the impacts of trampling on trails, to inform the development of appropriate management strategies [14]. Several authors [23–25] identified environmental changes associated with trails, including soil compaction, vegetation removal, modification of existing drainage patterns by topsoil removal and the modification of microtopography. In turn, these changes influence the microclimate. In addition, erosive processes can often be observed on trails, including splash erosion, rill erosion and even gullies. These erosive processes can also impair the quality of user experience and increase the risks of accidents.

Studies on the impact of trampling on trails are essential for soil quality analysis, especially in protected areas (PAs). It is important that these areas are properly managed, in order to conserve forest systems. We investigated two trails in PAs that form ecological corridors and receive intense and regular use.

## 2. Materials and Methods

### 2.1. Study Area

Two PAs were selected from the 'Atlantic Forest Biosphere Reserve', which is the first Brazilian unit within the 'World Biosphere Reserves Network'. It is the largest forest biosphere reserve in the world. The main objectives are the conservation of the biome through implementation of a continuous ecological corridor of coastal Atlantic Forest, linking existing forest [13].

The PAs are also part of the '*Serra do Mar* Biodiversity Corridor.' The two PAs are situated in an area of great touristic area appeal, especially for coastal attractions and waterfalls. They have two different administrations, one Federal (*Serra da Bocaina* National Park (SBNP)), located in *Rio de Janeiro* State and another (*Serra do Mar* State Park (SMSP)), located in *São Paulo* State (Figure 1).

It is important to assess soil degradation on trails and adjacent areas to inform planning and management in PAs. Therefore, this research evaluates soil quality on two trails, SBNP and SMSP. Investigations assessed the trampling impacts caused by visitors. The information was submitted in to the Administrations of both PAs, to assist in decision-making processes regarding land management.

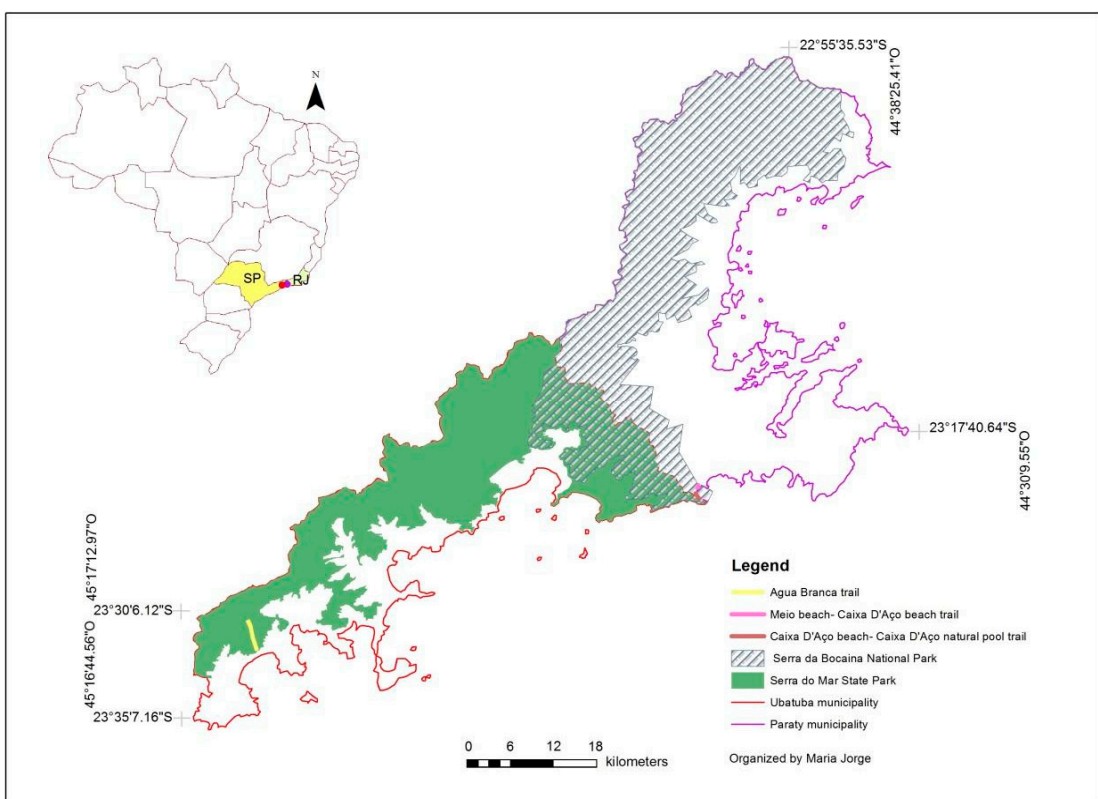

**Figure 1.** Study area. Organized by Maria Jorge.

### 2.1.1. Serra da *Bocaina* National Park

SBNP is composed of granites and gneisses [27,28] and the park coast forms part of a 'scarp border' relief unit, with local slopes >27° [29]. Local soils have clayey to silty textures, corresponding to associations of Inceptisols, with moderately developed to prominent A horizons [30]. Coastal vegetation is mainly secondary dense forest, at medium to advanced stages of recovery [30–32].

In the Köppen climatic classification system, SBNP is characterized by a humid tropical climate. Annual rainfall is ≤ 2200 mm, with mean values of ≤ 1700 mm [30,31]. Local climate is influenced by relief compartmentalization and topography, which cause variable spatial-temporal precipitation and temperature patterns.

Trindade village, where the studied trails are located, is within the *Serra da Bocaina* National Park boundaries and is a major tourist destination [31,32]. In addition, due to the aesthetic quality of the tourist attractions, several trails are intensively used by tourists, particularly the two that give access to *Caixa D'Aço* natural pool. This is the trail from *Meio* beach to *Caixa D'Aço* beach (MCB), which is ~190 m long, and the trail from *Caixa D'Aço* beach to *Caixa D'Aço* natural pool (CNP), which is ~465 m long (Figure 2). The expansion of tourism has caused several land degradation processes, including erosion, landslides and vegetation degradation. Although there is no official record of the number of visitors using the SBNP trails, managers estimated that during the 2018 local carnival, ~10,000 visitors used the trails.

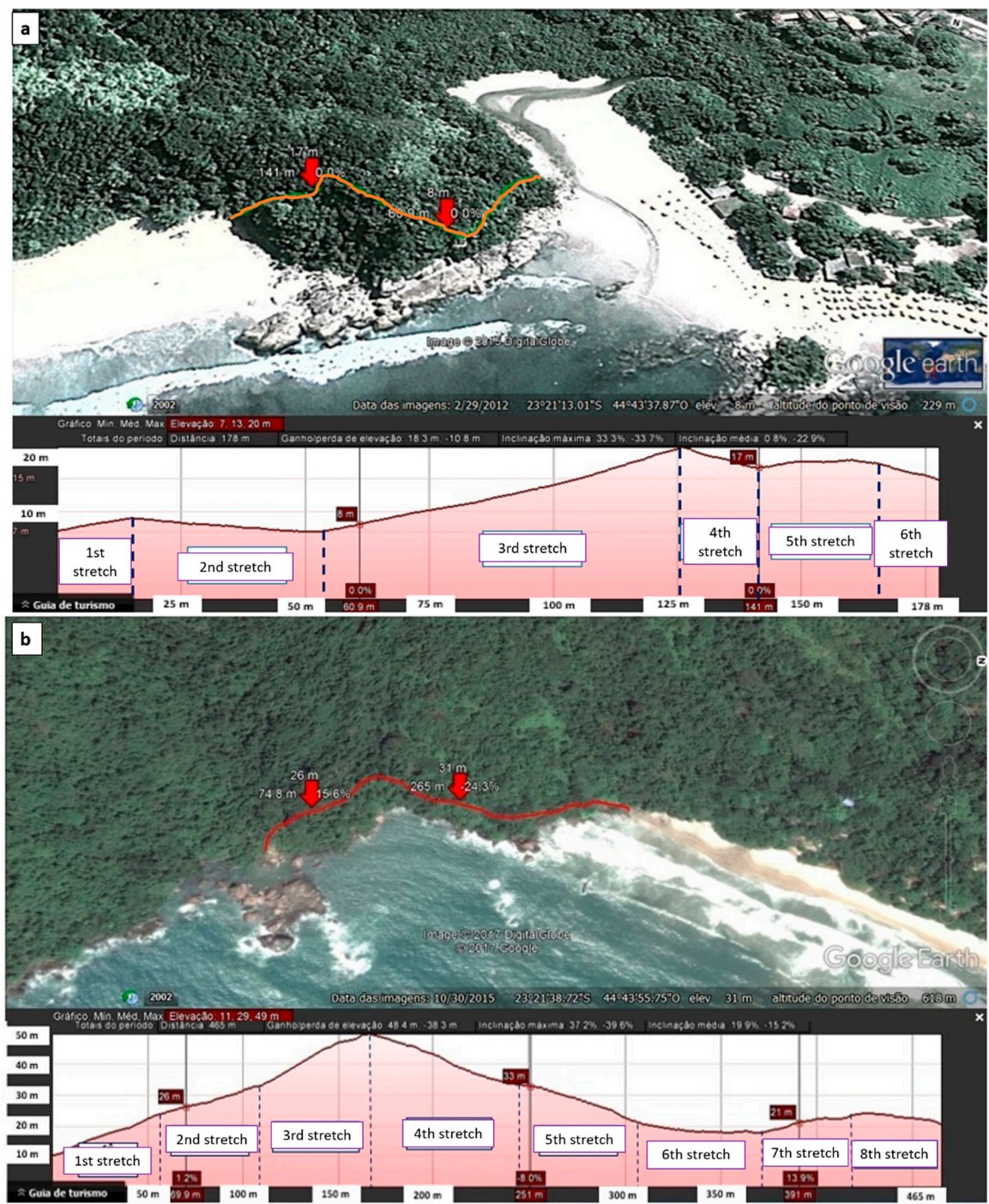

**Figure 2.** Elevation profile, location of 'erosion bridges' and division of MCB (**a**) and CNP (**b**) trails into sections based on topography. Adapted by L. Rangel from Google Earth. Image Source: Digital Globe.

### 2.1.2. *Serra do Mar* State Park

*Serra do Mar* State Park is composed of granites and gneisses, which overlie the Proterozoic-Eopaleozoic and Mesozoic crystalline basement. Cenozoic sediments are distributed throughout the coastal plain [33]. The main soil type is Inceptisols, with substantial areas of Oxisols, Entisols and Histosols. The Inceptisols are associated with hilly to mountainous relief and the fluvial plains. Entisols mainly occur on the steep slopes of the *Serra do Mar* mountain range [34].

The Köppen climatic classification is tropical maritime, being warm and humid, with a mean annual temperature of 19 °C. Maximum annual rainfall is ~4000 mm, with a mean of ~2500 mm [35,36].

These high rainfall amounts are caused by humid tropical maritime airstreams advected from the Atlantic Ocean, which condense as they reach the orographic barrier of the *Serra do Mar* mountain range [36]. These physical conditions promote the development of Atlantic Forest, which is one of the 34 world 'hotspots' for conservation, due to its high biodiversity and many indigenous species of flora and fauna [37].

### 2.2. Land Degradation in Serra da Bocaina National Park

To measure soil microtopography, erosion stakes were installed in cross-sections of the trail bed, from one border to another [38]. Four points were monitored (EB1, EB2, EB3, EB4), two in each SBNP trail. These were eroded areas of the trails. Measurements were taken in June and October 2016 and were repeated in February and June 2017. Four cross-profiles were measured, two on both the MCB and CNP trails.

The 'erosion bridge' was developed by Shakesby [39] and has been adapted for trails by several authors [13,40]. For the preparation of the profile, 50 cm long wooden stakes are used for levelling. The stakes remain at the measurement point, so that there are no levelling changes between the surveys. In addition, 2 m battens (erosion bridge) and a 1 m iron measuring rod are used. The bridge has 100 holes (analysis points), distributed at 2 cm intervals. To install the erosion bridge it is necessary to insert the two stakes into the edges of the selected cross-section. Subsequently, a spirit-level is used to level the erosion bridge, thus maintaining greater measurement accuracy. The values of each analysis point were measured using a measuring tape, after levelling the bridge (Figure 3). Chronosequences of topographic evolution were collated using EXCEL software. Analysis of graphs enables the identification of zones of soil erosion, sediment accumulation and intense trampling and to estimate soil erosion rates.

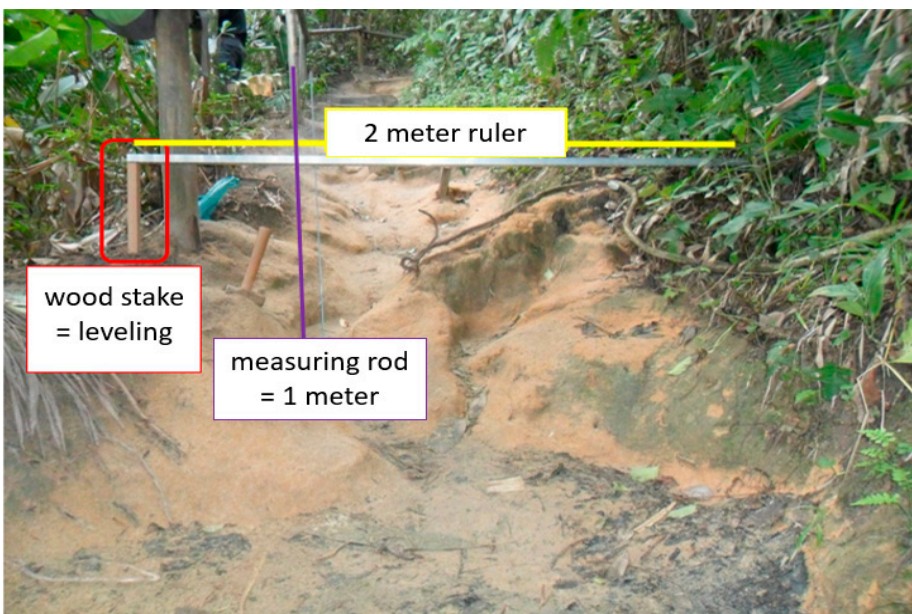

**Figure 3.** Measuring soil microtopography using the erosion bridge on *Caixa D'Aço* pool trails. Photo: L. Rangel (2016).

### 2.3. Land Degradation in Serra do Mar State Park

Disturbed and undisturbed soil samples were collected from 0–10 cm depth at four different sites on *Água Branca* trail, which is 4.5 km long (Figure 4). The samples were collected on the trail itself (TR) and the adjacent talus slope (TA). The undisturbed soil samples were used to determine bulk density. The disturbed samples were air-dried and soil texture, mineral density, SOM content, porosity and

pH were measured. All soil properties were determined using EMBRAPA protocols [41]. The Soil Taxonomy system of the United States Department of Agriculture was used [42].

Bulk density uses a 100 cm$^3$ steel cylinder, in which soil samples are collected [41]. In the laboratory, samples are removed from the cylinder and oven-dried for 24 h at 105 °C. Samples are allowed to cool to room temperature and then sample weight is divided by 100, to determine bulk density (g/cm$^3$).

Soil texture was determined using 20 g of soil, to which 10 mL of dispersant and 100 mL of distilled water were added [41]. After that, the mixture was mechanically shaken for 15 min and washed through a 0.053 mm sieve, which retained the sand fraction. The silt + clay fractions were filtered into a 1000 mL beaker. After a set time, 50 mL of water and soil were pipetted from the beaker and the clay fraction was collected. Sampling times are stipulated by EMBRAPA and these vary with sample temperature, Samples were then oven-dried for 24 h at 105 °C. Finally, samples were sieved through a 0.20 mm sieve, to separate the fine and coarse sand. Collated data were then plotted on the textural triangle template.

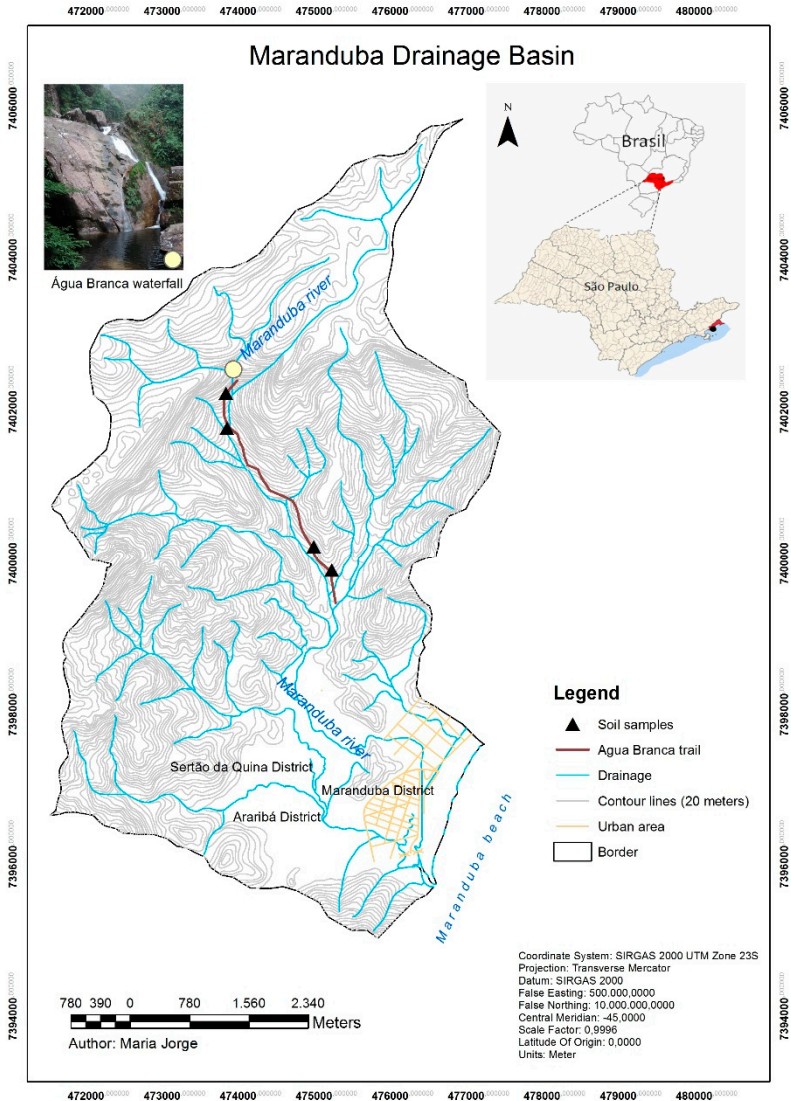

**Figure 4.** *Água Branca* trail in Maranduba drainage basin, Ubatuba Municipality. Figure prepared by Maria Jorge.

SOM was determined by wet chemical oxidation with potassium dichromate and three drops of diphenylamine. The analytical protocol involves multiple stages and SOM is determined by

titration [41]. SOM is reported as g/kg (i.e., C (g/kg) x 1.724). For pH analysis, 10 g of soil was added to 25 mL of distilled water. The mixture was shaken and allowed to stand for one h. Soil pH was measured using a calibrated pH meter [41]. Soil porosity (i.e., the total volume of voids within the soil occupied by water and/or air) is directly related to soil density and compaction and was calculated using EMBRAPA protocols [41].

## 3. Results

Analysis of soil microtopography allowed quantification of erosion and sediment deposition on the trails. Four transversal profiles were measured on the SBNP trails. However, due to heavy rainfall, it was not possible to measure soil microtopography on EB3 and EB4 in June 2017.

In the first erosion bridge measurements (EB1) there was a notable rill (depth 16 cm, width ~21 cm). Between October 2016–September 2017 the rill incised by a further ≤10 cm and some C horizon soils were exhumed (Figure 5).

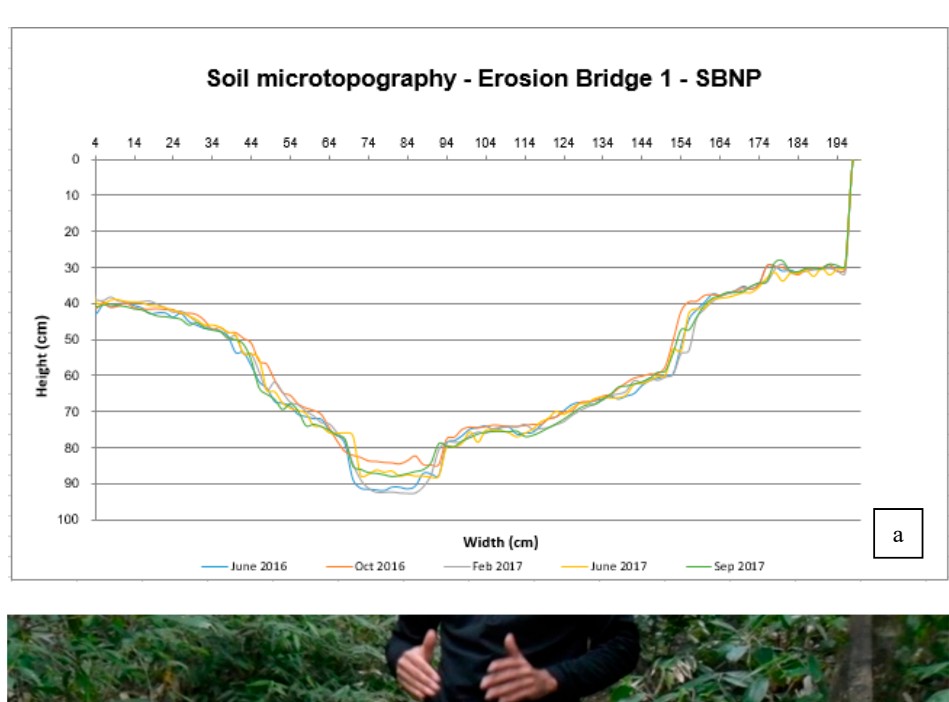

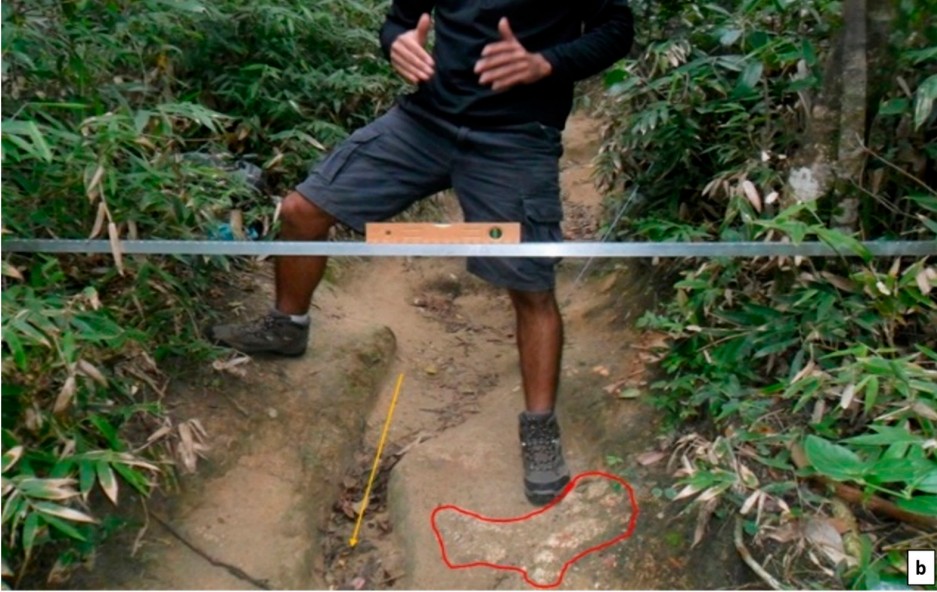

**Figure 5.** (**a**) Soil microtopographic analysis between June 2016 and September 2017 on EB1. (**b**) The yellow arrow indicates a rill containing accumulated litter. The red highlight indicates the onset of the exposure of the C horizon. Photo: L. Rangel (2017).

Figure 6 shows two rills on EB2, one ~20 cm deep and 24 cm wide and another ~13 cm deep and 28 cm wide. The estimated soil loss due to erosion on EB2 was 0.28 m² in June 2019 and 0.29 m² in October 2019. On EB2, between January and February 2017, a small landslide occurred on the upper slope of the trail, which became a shortcut for walkers.

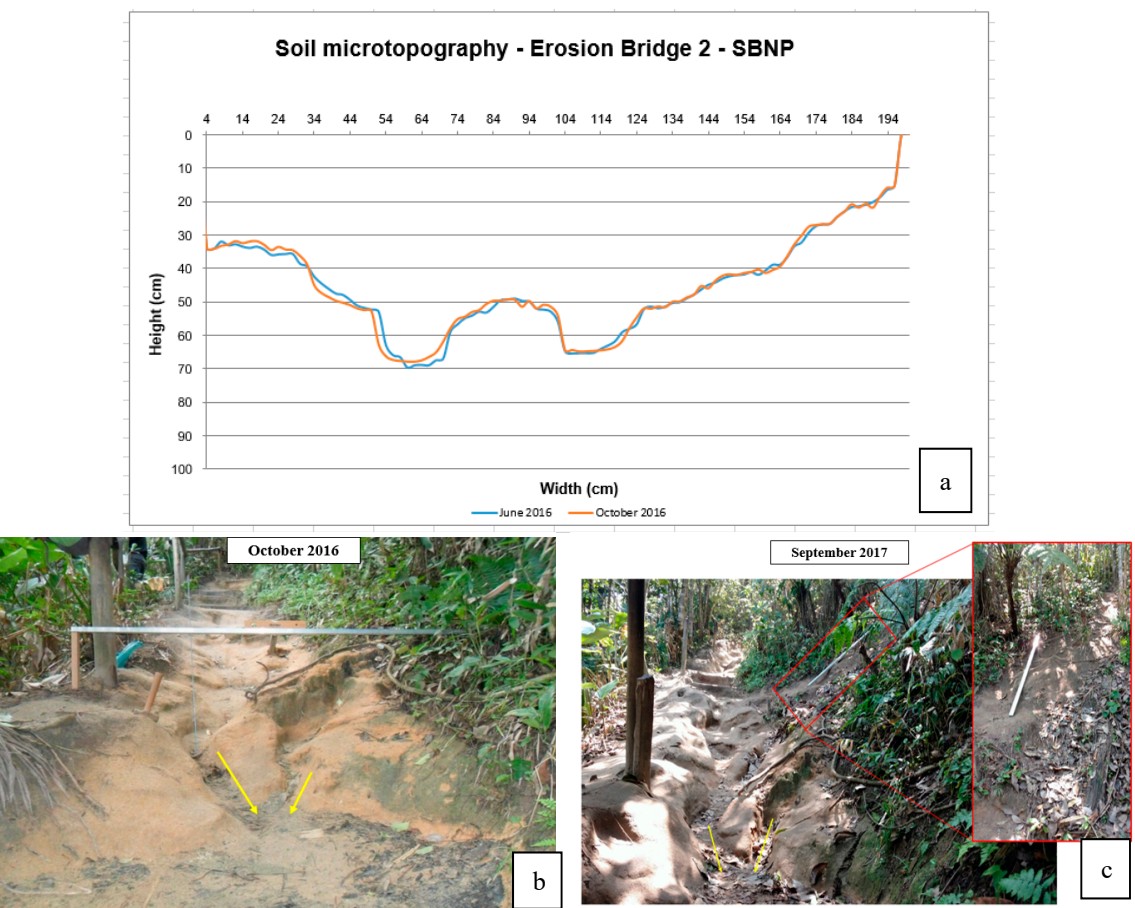

**Figure 6.** (**a**) Soil microtopographic analysis between June 2016 and October 2016 on EB2. (**b**) The yellow arrows indicate rills with litter accumulation in October 2016. (**c**) Landslide (detail in red) in September 2017. Photos: L. Rangel (2016 and 2017).

Analysing data from the third erosion bridge survey (EB3), it is possible to identify a rill partially infilled with leaf litter. Between October 2016–September 2017 the rill incised by a further ≤5 cm (Figure 7). On EB4, between October 2016–February 2017, the rill incised a further ≤12 cm. The effects of trampling and runoff are evident in the formation of three rills (Figure 8).

By analysing the graphs, it is possible to estimate the area of soil eroded from the cross-sections (Table 1). On SMSP, soil bulk density showed little variation between talus and trail ground, with means of 1.08 and 1.19 g/cm³. Values between 1.0–1.4 g/cm³ are considered medium [2]. Inversely, porosity values were high for TA and a little lower on TR (Table 2). High values promote high water infiltration rates into the soil. Soil pH values were very low for both TA and TR, which is indicative of intense leaching processes.

The textural classification of all soil samples were sandy-loam; a texture which is very erodible [10,13,28,43]. Nevertheless, some characteristics, such as the presence of vegetation on the trail edge and litter on the trail ground, have influenced the high SOM contents, which may have partially protected soils from erosion.

Roots were evident on the four sites, but water storage was not. Exposed roots appear on the trail surfaces due to the high slope angle and the presence of Entisols. Generally, Entisols are very shallow

and stony and do not have B horizons. Entisols and Inceptisols on steep slope angles are at risk of high erosion rates. The main geomorphological processes on *Água Branca* trail are mass movements, which decrease of trail width due to the collapse of trail edges. Generally, the trail is <1.30 m wide, and the width of the adjacent trampled area is ~0.40 m. Roots and litter on trails can provide habitats for snakes to hide, thus posing a potential hazard to walkers.

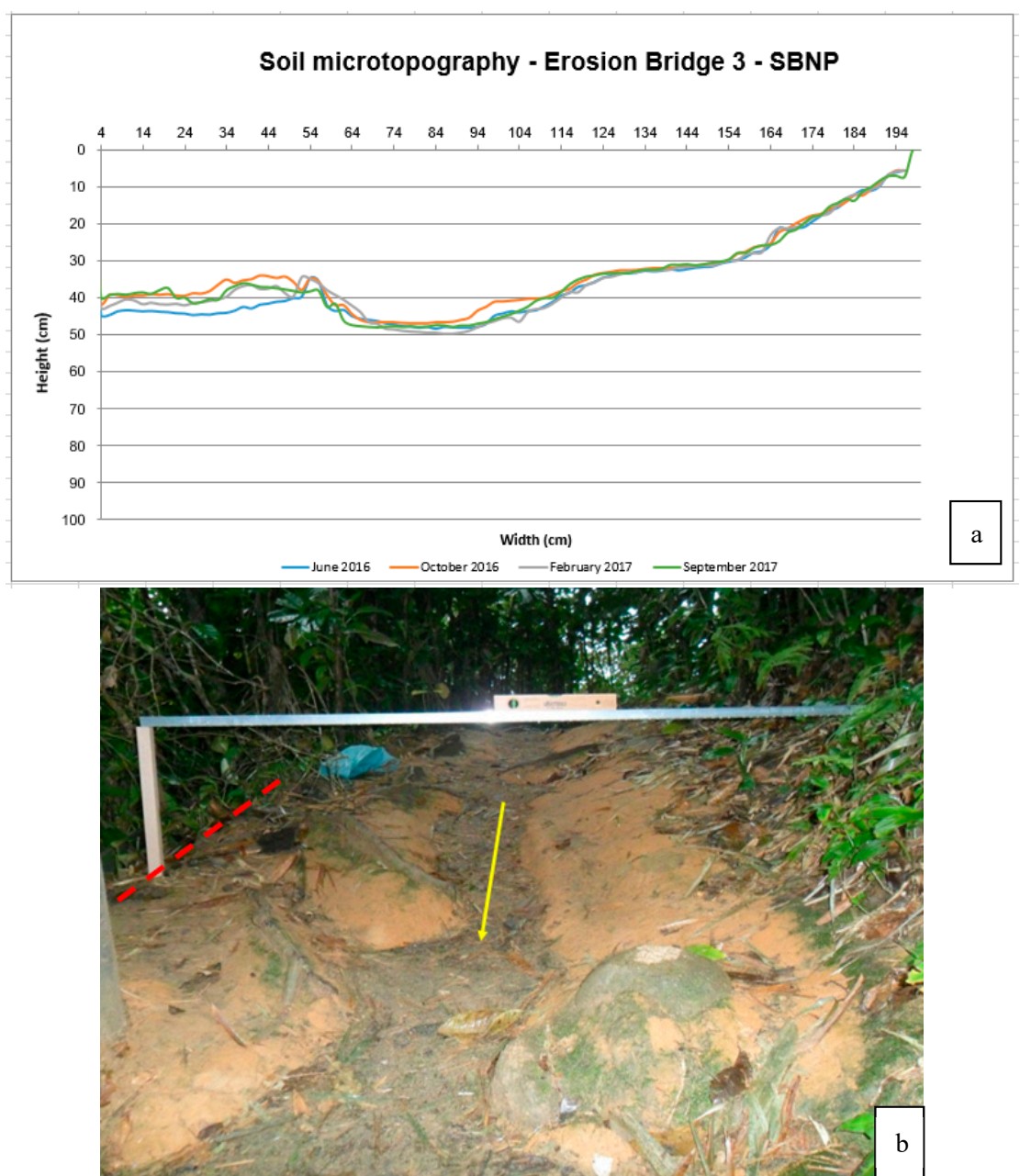

**Figure 7.** (**a**) Soil microtopographic analysis between June 2016 and September 2017 on EB3. (**b**) The yellow arrow indicates a rill with litter accumulation and the red line indicates erosion on the edge of the trail. Photo: L. Rangel (2017).

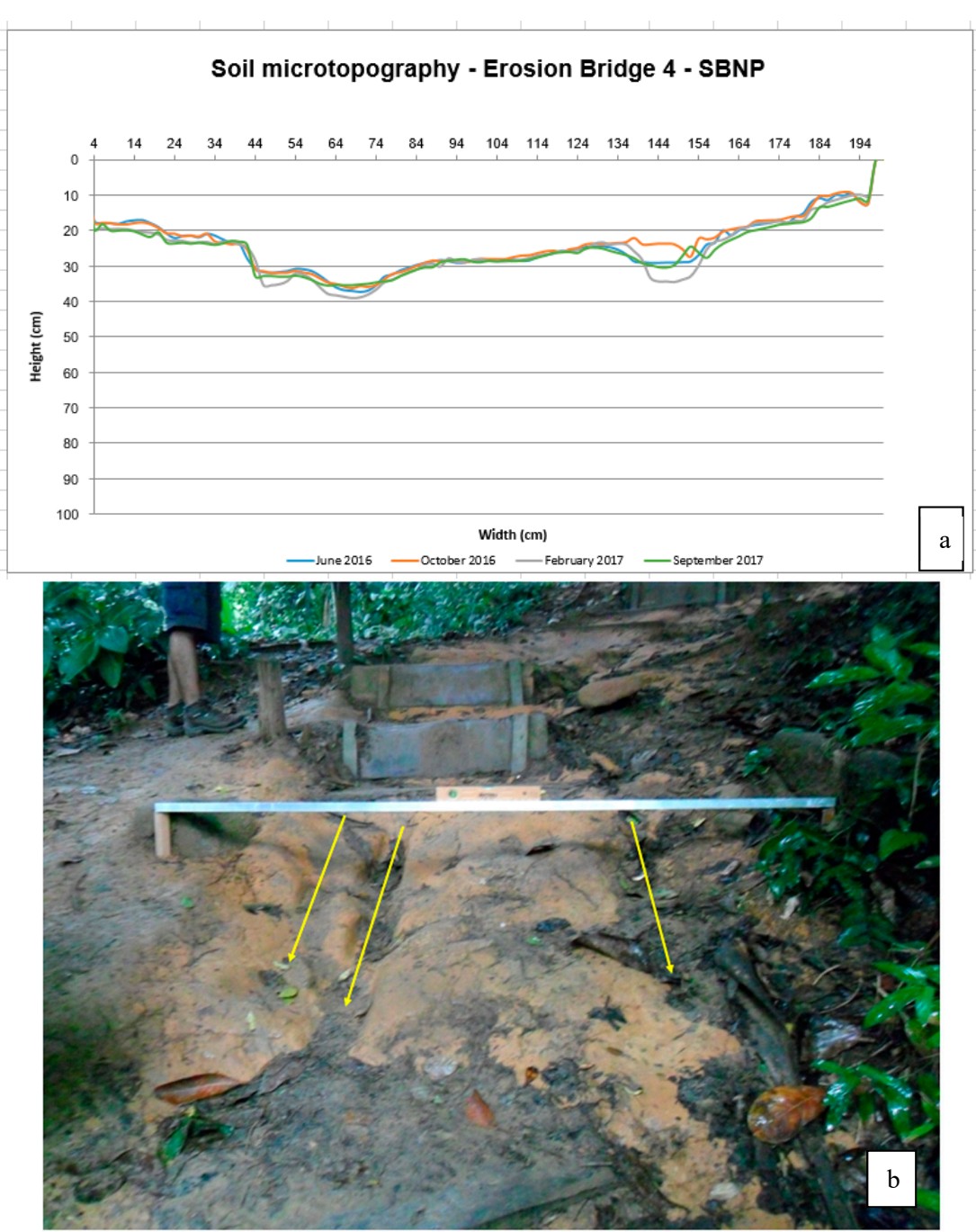

**Figure 8.** (**a**) Soil microtopographic analysis between June 2016 and September 2017 on EB4. (**b**) The yellow arrows indicate rills partially infilled by plant litter. Photo L. Rangel (2017).

**Table 1.** Estimated area ($m^2$) of eroded soils on the SBNP trails.

|  | June 2016 | October 2016 | February 2017 | June 2017 | September 2017 |
|---|---|---|---|---|---|
| EB1 | 0.39858 | 0.35816 | 0.39384 | 0.38342 | 0.38948 |
| EB2 | 0.27912 | 0.27823 | - | - | - |
| EB3 | 0.21172 | 0.16538 | 0.19456 | - | 0.18428 |
| EB4 | 0.20616 | 0.19352 | 0.22512 | - | 0.22466 |

**Table 2.** Soil properties on trails in *Serra do Mar* State Park. Coarse sand (2.0–0.2 mm), fine sand (0.2–0.053 mm), silt (0.053–0.002 mm) and clay (<0.002 mm) (source: 41).

| | **Mean Bulk Density (g/cm$^3$)** | | | | | |
|---|---|---|---|---|---|---|
| Trail *Agua Branca* | Talus (TA) | | | | Trail ground (TR) | |
| | 1.08 | | | | 1.19 | |
| | **Mean porosity (%)** | | | | | |
| Trail *Agua Branca* | Talus (TA) | | | | Trail ground (TR) | |
| | 55.51 | | | | 49.78 | |
| | **Mean soil organic matter content (%)** | | | | | |
| Trail *Agua Branca* | Talus (TA) | | | | Trail ground (TR) | |
| | 7.34 | | | | 8.55 | |
| | **Mean pH** | | | | | |
| Trail *Agua Branca* | Talus (TA) | | | | Trail ground (TR) | |
| | 3.71 | | | | 3.69 | |
| | **Mean texture (%)** | | | | | |
| Trail *Agua Branca* | Mean | Coarse sand | Fine sand | Silt | Clay | Classification |
| | Talus (TA) | 45.05 | 13.39 | 26.73 | 14.83 | Sandy loam |
| | Trail ground (TR) | 50.05 | 11.99 | 26.87 | 11.09 | Sandy loam |

## 4. Discussion

On the SBNP trails (EB1), the quantity of material deposited within rills can be associated with litter accumulation, which varies according to the production and decomposition cycle of the organic material and the rainfall regime [13]. Soil erosion probably occurred due to a combination of intense rainfall, local steep slopes and the erodible sandy texture.

On EB2 it was only possible to conduct two monitoring sessions, since in February 2017 one of the cuttings that served as a levelling base was removed by a landslide that occurred in the upper slope of the trail and the other was eroded as a result of this slippage. These processes are associated with the high local rainfall amounts. Between 1 January–3 February 2017 (the dates of the surveys), rainfall at Paraty Weather Station was 485 mm. During 3-h on 10 January 2017 ~101 mm of rain fell [44].

Silva and Castro [40] also found deep rills on the *Perigoso* beach trail (*Rio de Janeiro* State). The first rill was 20–30 cm wide and 20 cm deep. The second rill was 50–60 cm wide and 30–35 cm deep. The authors highlighted the negative influences of soil erosion features on the visitor experience. The same was found in the present study, as several visitors had difficulties walking over the eroded areas.

In EB3, the rill (~50 cm wide, ~15 cm deep) shows the preferential path of runoff and visitors, who further compact the soil by trampling. On EB4, the presence of three rills may be associated with steep microslopes.

Soil microtopography is changing due to soil erosion. Observations suggest changes can be attributed to a combination of trampling intensity, runoff being funneled into rills and high soil erodibility. Furthermore, the absence of vegetation and organic matter on the trails exposes soil to erosive rainfall. On the *Água Branca* trail, SOM content is higher on the trail ground than the talus, because of leaves falling from adjacent trees.

Erosion processes, such as rills and mass movements, occur on many parts of the trail. Together with the lack of information and handrails, trails can be dangerous for users and diminish interest in visiting the area. These trails are mainly used by tourists to access waterfalls and for bird watching. Therefore, local authorities should ensure the safety of walkers, which will help to promote local economic development.

The assessment of trail impacts due to erosion processes shows the importance of preparing and implementing a maintenance plan for the *Água Branca* trail. Therefore, some measures are advocated to improve safety and to contribute towards the development of geotourism and natural resource conservation. Considering suggestions by Jorge et al. [10] and Rangel and Guerra [13], the following measures are proposed:

1. Construction of steps with handrails, where the trail presents risks to walkers.
2. Construction of drainage on the trail edge, diverting water flow in selected places, and consequently, arresting erosion processes.
3. Plant suitable vegetation on trail edges, especially where the risk of soil erosion is high.
4. Introduce litter to the trail surfaces, as a means of buffering soil compaction and protecting the soil surface against direct rainfall impact.
5. Establish the best trail route, because there are many bifurcations created by walkers, which can accelerate erosion and make walking difficult.
6. PROMATA (the local environmental NGO) should co-operate with the State Park administration and the local community to agree appropriate and viable strategies to improve the trails.
7. Establish information boards along the trail, including interpretative posters, in a way that the geosites are clearly explained and in terms that can be understood by non-specialists. Information should include the trail details and the locally rich and diverse geology, soils, rock outcrops, fauna and flora. Currently, there is no information provision.

## 5. Conclusions

The deficient trail management has negative impacts on soil quality. Therefore, it is necessary to rehabilitate degraded areas. Appropriate strategies include bioengineering techniques, such as the application of geotextiles; adding organic matter to soils, and the installation of hydrological structures (e.g., check dams and drainage barriers). Soil microtopographic analysis proved very effective in measuring soil erosion and sediment accumulation.

The *Água Branca* trail does not undergo as much human trampling as the Serra da *Bocaina* National Park trail. However, the combination of an erodible soil texture (sandy loam), steep slopes and erosive rainfall has caused much soil degradation on the trail, mainly in the form of soil erosion and mass movements. The trails need improved management by the State Park authorities. In turn, by attracting geotourists, better management could assist the development of sustainable tourism and associated economic benefits.

**Author Contributions:** Data curation, L.R. and M.d.C.J.; Formal analysis, L.R. and M.d.C.J.; Funding acquisition, A.G.; Investigation, L.R. and M.d.C.J.; Methodology, L.R. and M.d.C.J.; Project administration, A.G.; Supervision, A.G. and M.F.; Validation, M.F.; Visualization, L.R., M.d.C.J., A.G. and M.F.; Writing of the original draft, L.R. and M.d.C.J.; Writing review and editing, L.R., A.G. and M.F.

**Funding:** This research was funded by FAPERJ (*Rio de Janeiro* Research Council) Grant Number E-26/203.309/2017, CNPq (the Brazilian Research Council), and CAPES (Coordination of Personnel of University Teaching) through PhD grants.

**Acknowledgments:** The authors thank FAPERJ, CNPq and CAPES for financial support and PROMATA (Program for the Conservation of the Atlantic Forest in Ubatuba Municipality) for their logistical support.

**Conflicts of Interest:** The authors declare no conflict of interest.

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
