# Peer review of "Soil Erosion and Land Degradation on Trail Systems in Mountainous Areas: Two Case Studies from South-East Brazil"

_soilsystems, doi:10.3390/soilsystems3030056_

Round 1

Reviewer 1 Report

This is a very interesting manuscript. Field investigation and sample measurements were conducted to reveal the erosion processes of the trails in reserve areas. I recommend it be accepted, however, I think some information should be added before be accepted, including the rainfall information in the study area and the number of people that ran over the trails. These data are critically important to influence erosion processes of the trails

Author Response

All suggested revisions were made as highlighted in the attached file

Reviewer 2 Report

Introduction - can be improved  (develop the possible effects of rails on environmental and productive soil functions)

Material and Methods - refer to the parameters used in the evaluation of soil quality.  Improve the reading of Figure 1. Put the meaning of the acronyms used.

Results and Discussion - improve the reading of Table 1. In the results and discussion, you must explore the data obtained that allowed to evaluate the soil quality; you only discussed the erosion results. The evaluation of soil quality is one of the objectives proposed and has not been explored. 

Conclusions - some conclusions are not supported by the results.

Author Response

All suggested revisions were made as highlighted in the attached file.

Reviewer 3 Report

This is a study of soil erosion on two Brazilian trails (abbreviated SBNP and SMSP). Both of them are situated in protected areas but administered by different park authorities (one federal and one state). The first trail (SBNP) has two segments—MCB and CNP. Four erosion bridges were set up on the trail to monitor the change of ground surface. The second trail does not have erosion bridges. The authors only took soil samples from the second trail to measure the bulk density, porosity, organic content, and pH of the soil. Based on the findings, the authors recommended measures to protect visitors and residents using the trails and to reduce soil erosion. The manuscript is well organized and written, but the content can be substantially improved.

First of all, the manuscript is weak and lacking technical content related to soil erosion. For trail #1 (SBNP), perhaps the authors should consider adding more quantitative data to the results in addition to showing the trail transects and their cross-sectional areas. For example, what are the average erosion depths of the trail transects (overall and per year) at the locations of the four erosion bridges? What are the equivalent amounts of soil erosion in terms of metric tons of soil per year per hectare (t/ha/yr)? Could the authors compare the trail erosion with the erosion of nearby areas (not used for trails) to isolate the effect of human trampling? Are there similar studies in the literature? How do the results of this study compare with the results in other studies? Could the authors correlate the trail erosion (depth) with the number of visitors and residents using the trail (or at least the number of persons visiting the park)?

In addition, the authors seem to use the cross-sectional areas (areas hatched by green lines as shown in Figures 5-8) to represent the amounts of soil loss (lines 178, 190, 203 and 214), but they are not correct. For example, the authors only made two measurements at erosion bridge #2. They were in June 2016 and October 2017, respectively. But the authors stated that:

"The area (green hatching) estimated of soil loss due to erosion in PE 2 is 0.28 m² in June and 0.29 m² in October."

This is obviously a false statement. Because the authors did not start their monitoring before the soil erosion took place (a level ground), the only thing that they could infer from the measurements is that the cross-sectional area at erosion bridge #2 changed by 0.1 m² from June 2016 to October 2017. Incorrect inference and erroneous interpretations similar to the above statement can be found in the manuscript for the other three erosion bridges.

For trail #2, only very basic soil properties such as bulk density, porosity, organic content, and pH were measured. No erosion measurement was made and less than half a page of text was devoted to this trail in the results section (lines 223-245). Hence, it hardly constitutes a case study of "soil erosion and land degradation" as specified by the title of the manuscript. With no direct measurements of soil erosion, what is the purpose of including the second trail in the manuscript? What do the authors intend to show? What is the relationship of the second trail to the first trail?

Additionally, the authors used a Brazilian soil classification system (EMBRAPA) to classify soil samples of trail #2. The sieve sizes (lines 156-159) of the Brazilian system seem to be different from those of the commonly used soil classification systems such as the USDA, Unified, and AASHTO classification systems. The authors need to provide more details of the Brazilian system and its textual triangle.

Finally, although the manuscript is well written, some explanations are unclear and the English could still use a good proofreading. For example, the sentence starting from line 177 is obviously incomplete. There are others like that in the manuscript.

More detailed comments:

- Lines 47-48: can people live in abiotic areas?

- Figure 1 is too small and the fonts are not legible.

- Figure 2 is also too small to read. The green line used in the upper figure is very difficult to see.

- Lines 82 and 89: misspelling of SBNP.

- Figure 6: where is the purple line?

- Table 1: extra "4" at the beginning of line 6.

Author Response

(The authors gave the same response as above.)

Reviewer 4 Report

Suggest going through the paper once again and giving it one more extensive edit for English language and spelling/grammar checks. Most of the manuscript is readable, but in some places it is "clunky", including to the Conclusions. Check for a number of run-on or incomplete sentences. 

For the Figures, particularly 2, 5, 6, 7, and 8, I suggest making these higher quality. Where more than 1 panel is shown that includes numerical data, it is not also clear and readable, which destroys the purpose of having a figure (especially Figure 2, this reader cannot see the inserted data to understand the aerial photograph even with zooming in). Additionally, for the Excel panels, I like the concept but the design is not user friendly- unnecessary lines should be removed as well as border on graphs and pictures. Panel letters to delineate each part of the figure should not obscure or block any part of the display. Importantly, the main figures have a reference which is NOT in the style of the journal, and it is unclear if these figures are being reprinted or are original. 

Author Response

(The authors gave the same response as above.)

Round 2

Reviewer 2 Report

The article is acceptable to published

Author Response

The introduction has been improved (lines 64-71)

Figure 1. Map has been improved, with better font and explained acronyms 

Reviewer 3 Report

The authors did not provide a point-by-point response to the questions raised in my previous review comments. From the revised manuscript itself, it seems that the authors ignored my comments on adding more quantitative data, calculating the average depths of trail transects, calculating the equivalent amounts of soil erosion, isolating the effect of human trampling, explaining the Brazilian soil classification system, etc. without any explanation. I am afraid that I must insist the authors to provide a point-by-point response to my earlier comments.

In addition, I notice that the authors still used the cross-sectional areas to represent the amounts of soil loss (lines 203-204) or used the wrong unit of cm (lines 190, 213, and 220). At least the numbers need to be converted to the average depths of trail transects between the starting and ending dates of data collection. The new Table 1 on p. 11 can be used for this purpose, but the last column needs to be changed to the difference in cross-sectional areas per unit width (of trail transect) per unit time.

Author Response

The two trails are used to exemplify different methodologies for analysis of erosive processes in protected areas located in dense forest. For the trail # 1 more data have been included (Table 1), and the green hatching graphs area has been removed. We have monitored, based on the erosion bridges, the depth evolution, of the eroded areas, from the methodology proposed by (Ferreira, 1996) and Silva and Castro (2015). Therefore, the erosion bridge monitors the erosion depth evolution that is why we have used centimeter units, to indicate the rills depth, and m² to indicate the area eroded. This methodology does not aim at estimating the amount of eroded soil, at each cross section surveyed per year, because on the trail ground there is litter deposition, which interferes on the soil loss calculation. Therefore, the suggestion for the last column of Table 1 (difference in cross-sectional areas per unit time) cannot be attended, because these data would not be accurate. When we make the measurements, we consider both trampling, and surface flow erosion (lines 224-225). It is not possible to compare the trails erosion with other close areas, because this is a Protected Area, and consequently, it is covered by dense tropical forest. According to the available literature, regarding soil erosion on trails, there are few data, mainly using the erosion bridge methodology. We know that there is only one study, which has applied this methodology on trails, therefore this article is pioneer. Therefore, the comparison between this present research work and the other has been carried out (lines 274-279).

Soil classification proposed by EMBRAPA (2013), used in this article, is the same as U.S. Soil Taxonomy. Step by step of the methodology has been detailed on this article. The sieve sizes to separate coarse from fine sand is the same used on other international methodologies, being the same sediment texture classification (lines 167-168).